# Design, Synthesis and Bioactivity of Novel Pyrimidine Sulfonate Esters Containing Thioether Moiety

**DOI:** 10.3390/ijms24054691

**Published:** 2023-02-28

**Authors:** Changkun Li, Youhua Liu, Xiaoli Ren, Yanni Tan, Linhong Jin, Xia Zhou

**Affiliations:** Key Laboratory Breeding Base of Green Pesticide and Agricultural Bioengineering, Key Laboratory of Green Pesticide and Agricultural Bioengineering, Ministry of Education, Guizhou University, Guiyang 550025, China

**Keywords:** pyrimidine, sulfonate, synthesis, antibacterial activity, defense enzyme

## Abstract

Pesticides play an important role in crop disease and pest control. However, their irrational use leads to the emergence of drug resistance. Therefore, it is necessary to search for new pesticide-lead compounds with new structures. We designed and synthesized 33 novel pyrimidine derivatives containing sulfonate groups and evaluated their antibacterial and insecticidal activities. Results: Most of the synthesized compounds showed good antibacterial activity against *Xanthomonas oryzae* pv. *Oryzae* (*Xoo*), *Xanthomonas axonopodis* pv. *Citri* (*Xac*), *Pseudomonas syringae* pv. *actinidiae* (*Psa*) and *Ralstonia solanacearum* (Rs), and certain insecticidal activity. **A_5_**, **A_31_** and **A_33_** showed strong antibacterial activity against *Xoo*, with EC_50_ values of 4.24, 6.77 and 9.35 μg/mL, respectively. Compounds **A_1_**, **A_3_**, **A_5_** and **A_33_** showed remarkable activity against *Xac* (EC_50_ was 79.02, 82.28, 70.80 and 44.11 μg/mL, respectively). In addition, **A_5_** could significantly improve the defense enzyme (superoxide dismutase, peroxidase, phenylalanine ammonia-lyase and catalase) activity of plants against pathogens and thus improve the disease resistance of plants. Moreover, a few compounds also showed good insecticidal activity against *Plutella xylostella* and *Myzus persicae*. The results of this study provide insight into the development of new broad-spectrum pesticides.

## 1. Introduction

Plant pests and diseases are a major concern in agriculture. Rice bacterial blight, which is caused by *Xanthomonas Oryzae* pv. *Oryzae (Xoo*), can reduce rice production by more than 50% [1]. Citrus canker, caused by *Xanthomonas axonopodis* pv. *citri* (*Xac*), kiwifruit bacterial canker, caused by *Pseudomonas syringae* pv *actinidiae* (*Psa*) and tobacco bacterial wilt caused by *Ralstonia solanacearum* (Rs), can also have devastating effects on crops [2,3,4]. Currently, pesticides used to control these plant diseases mainly include commercial agents such as thiodiazole copper (TC), thiazolyl zinc (TZ) and Bismerthiazol (BT), but their frequent use over a long period of time has led to increased resistance and poor control results [5,6,7]. Therefore, there is an urgent need to develop a new broad-spectrum pesticide to control plant pests and disease infestations.

Pyrimidine derivatives have broad-spectrum biological activities, such as antibacterial [8], antifungal [9], insecticidal [10], herbicidal [11], antiviral [12,13], anti-inflammatory [14] and antitumor [15], and are of wide interest in the pharmaceutical and pesticide fields [16,17]. For example, commercial pyrimidine structures (e.g., mepanioyrim, pyrimethanil and ferimzone hydrazone) have been developed to control pests and diseases (Figure 1a). Therefore, the pyrimidine ring skeleton as a building block has proved to be a valuable guiding group for the development of new pesticides.

In addition, it is found that compounds containing aryl sulfonates have a wide range of biological activities such as antiviral [18], antibacterial [19], insecticidal [20] and anticancer [21] activities. Some commercial pesticides containing sulfonate structure are developed, for example, Bupirimate, Chlorfenson and Genitol (Figure 1b). New derivatives containing sulfonate structures are reported with the function of changing the physicochemical properties of the structures, promoting interaction with molecular targets, and improving bioavailability [22]. In addition, we reported that 1,3,4-oxadiazole sulfonate could provide excellent antibacterial activity [23]. Based on those prior works, we expect that the new structures may exhibit interesting bioactivities when the sulfonate ester is combined with the pyrimidine scaffold. With this respect, a series of novel pyrimidine derivatives containing sulfonate esters are designed and prepared based on the principle of splicing and replacement (Figure 1c). Hence in the present work, a series of novel compounds are synthesized by introducing sulfonate moiety to pyrimidine to discover new structures with potential antibacterial activity and insecticidal activity. The design and synthesis of the targets are depicted in Figure 1c and Figure 1, respectively.

## 2. Results and Discussion

### 2.1. Physicochemical Properties Analysis

The structure of derivative 4-[2-amino-6-(4-substituted phenyl)-pyrimidin-4-yl]benzenesulfonamide (**A_1_**–**A_33_**) was designed using ChemDraw Ultra 8.0 software, and a smile file was generated, that file was used to calculate the bioactivity score and physicochemical properties. Lipinski’s five rules for standard drug molecules are that no more than 5 H-bond donors, the molecular weight is not more than 500, the CLog P is not over 5 (or MLOGP is over 4.15), and the sum of N’s and O’s is not over 10. The results of the physicochemical analysis showed that the designed compounds all conformed to Lipinski’s rule of 5 (Table 1) [24,25]. The bioactivity score or drug-like score (A biological activity score of less than −5 indicates no activity, whereas a score greater than −5 indicates activity) corresponds to the active drug molecule, and the results show that the designed compounds (**A_1_**~**A_33_**) are located in the drug active molecule region (Table 2) [26].

### 2.2. Chemistry

#### 2.2.1. Synthesis

As described in Figure 1, the key intermediate (6-methyl-2-thioxo-2,3-dihydropyrimidin-4(1H)-one) methanol 1 is synthesized by cyclization using ethyl glycolate as starting material. Subsequently, intermediate 1 is converted into its corresponding thioether derivatives 2, 3 and 4 by thioetherification with CH_3_I, C_2_H_5_I and Benzyl chloride. Finally, the title compounds **A_1_**–**A_33_** are obtained by esterification with RSOOCl. The physical and chemical properties and the ^1^H NMR, ^13^C NMR and HRMS spectra of all target compounds are presented in Appendix A.

#### 2.2.2. Crystal Structure of Compound **A_10_**

Crystal data for **A_10_** are presented in Appendix A, while Figure 2 provides a perspective view based on the atomic labeling system (CCDC:2184366; these data are available free of charge from the Cambridge Crystallographic Data Centre). According to the structural analysis, **A_10_** has a U-shaped conformation, which is close to its binding pose.

### 2.3. In Vitro Antibacterial Activity

The antibacterial activities of the title compounds **A_1_**–**A_33_** against *Xoo*, *Xac*, *Psa* and Rs are determined by a turbidimetric assay in vitro. The results are shown in Figure 3 and Figure 4 (Appendix A). It can be seen most of the target compounds exhibited good anti-*Xoo* and anti-*Xac* activities in vitro. Among them, compounds **A_5_**, **A_30_** and **A_33_** showed excellent antibacterial activity against *Xoo* with EC_50_ values ranging from 4.24 to 9.35 μg/mL, which is remarkedly lower than exhibited TC (45.81 μg/mL) and BT (25.12 μg/mL). Moreover, compounds **A_1_**, **A_3_**, **A_5_** and **A_33_** possessed better antibacterial activity against *Xac* with EC_50_ values (44.11–82.28 μg/mL) lower than that of thiodiazole copper (130.49 μg/mL) (Figure 5).

### 2.4. Structure-Activity Relationship

For the anti-*Xoo* activity, the aromatic ring (86%) was greater than the pyridine ring (79%), the thiophene ring (75%), the aliphatic hydrocarbon (69%), the aliphatic hydrocarbon (45%) and the fused ring (28%) under the same sulfonate structure. When both sulfonates were connected by aromatic rings, the activity of substituents at different positions was different, and the parapace substitution **A_3_** (83%) was higher than the collar substitution **A_2_** (73%). The activity is not the same for the same position of different substituents –F (83%) > –NO_2_ (65%) > –Cl (62%), Br (62%) > –CN (58%) > –OCH_3_ (47%) > –CF_3_ (53%) > –C (CH_3_)_3_ (25%).

### 2.5. In Vivo Activity Analysis

With outstanding bactericidal activity in vitro, compound **A_5_** is further explored for its antibacterial potency in vivo against rice bacterial leaf blight via the leaf-cutting method. Bismerthiazol and thiodiazole copper served as positive controls for this investigation. All inoculated plants in 14 days exhibited blight symptoms with 100% morbidity.

At the concentration of 200 µg/mL, as shown in Figure 6 (Appendix A) and Figure 7, the title compound **A_5_** had selective protective and curative activities on Bacterial Leaf Blight (BLB) in vivo, with inhibition rates of 47.85 and 41.05%, respectively. The protective activity of **A_5_** is significantly better than TC (45.65%) but weaker than BT (57.02%). The curative activity is significantly better than TC (31.81%) but weaker than BT (52.54%). Additionally, as shown in Table 3 and Figure 8, the in vivo protective and curative activities of compound **A_10_** against citrus bacterial canker on citrus leaves are 27.68 and 26.86%, respectively, which are better than those of thiodiazole copper (26.24 and 21.03%).

### 2.6. Scanning Electron Microscopy (SEM) Characterization Analysis

As shown in Figure 9, when compound **A_5_** is at a concentration of 50 µg/mL, the bacterial cell is deformed, and part of the bacterial cell wall is slightly ruptured. When the compound **A_5_** concentration is increased to 100 µg/mL, most cell membranes are wrinkled and ruptured. Then observing the negative control group (A), these bacterial cells are round and smooth, without any breakage. Scanning electron microscopy images further demonstrated that compound **A_5_** has antibacterial activity against *Xoo*.

### 2.7. The Cytoplasmic Content of Xoo Treated with Compound ***A_5_***

The leakage of cytoplasmic contents of *Xoo* is determined by measuring the absorbance values (OD_260_ and OD_280_) at 260 and 280 nm, respectively. As shown in Figure 10, the OD_260_ and OD_280_ values of the groups treated with compound **A_5_** at a concentration of 21.2 μg/mL (5 equiv. EC_50_), 42.4 μg/mL (10 equiv. EC_50_), 42.4 μg/mL (10 equiv. EC_50_) and 84.8 μg/mL (20 equiv. EC_50_), respectively, are significantly increased compared to those of the blank control group and stabilized in approximately 48 h. The results are consistent with the morphological observation by SEM. Therefore, we speculate that there is a significant release of nucleic acid and proteins in the cells of *Xoo* treated with compound **A_5_**.

### 2.8. Changes in Xoo Defense Enzyme Activity by Compound ***A_5_***

The disease resistance of plants is closely related to defense-related enzyme activities. Based on the excellent protective activity against BLB, we compared some defense enzyme activities of rice after treatment with compound **A_5_**. The test results of CAT activity are indicated in Figure 11A. The **A_5_** + *Xoo* treatment group reached a peak CAT activity on the 5th day after infection with bacterial *Xoo*, and the activity data reached 317.86 U/g, while the CAT activity data of the CK treatment, CK + *Xoo* treatment, and TC + *Xoo* treatment were 214.89, 251.05, and 263.00 U/g, respectively. The CAT activity of the **A_5_** + *Xoo* treatment was 1.48, 1.27, and 1.21 times those of the CK treatment, CK + *Xoo* treatment, and TC + *Xoo* treatment, respectively. As shown in Figure 11C,D, the results show that the activities of POD and PAL in rice treated with compound **A_5_** showed a trend of increasing first and then decreasing. The **A_5_** + *Xoo* treatment had the highest POD activity on the 7th day after bacterial *Xoo* infection; the activity data reached 1010.96. U/g, while the POD activity data of the CK treatment, CK + *Xoo* treatment and TC + *Xoo* treatment were 585.50, 682.35 and 873.75 U/g, respectively. The POD activity of the **A_5_** + *Xoo* treatment was 1.73, 1.48 and 1.16 times those of the CK treatment, CK + *Xoo* treatment and TC + *Xoo* treatment, respectively. The **A_5_** + *Xoo* treatment had the highest PAL activity on the 7th day after bacterial infection; the activity data reached 3.85. U/g, while the PAL activity data of the CK treatment, CK + *Xoo* treatment and TC + *Xoo* treatment were 2.84, 3.04 and 3.43. U/g, respectively. The PAL activity of the **A_5_** + *Xoo* treatment was 1.36, 1.27 and 1.12 times those of the CK treatment, CK + *Xoo* treatment and TC + *Xoo* treatment, respectively. As depicted in Figure 11B, the activity of SOD in rice treated with compound **A_5_** showed a trend of increase at first and then decreased, reaching a peak on the 5th day. The activity data reached 149.89. U/g, while the SOD activity data of the CK treatment, CK + *Xoo* treatment and TC + *Xoo* treatment were 115.69, 142.65 and 122.57 U/g, respectively. The SOD activity of the **A_5_** + *Xoo* treatment was 1.29, 1.05 and 1.22 times those of the CK treatment, CK + *Xoo* treatment and TC + *Xoo* treatment, respectively. Compared with CK + *Xoo* treatment and CK treatment, the SOD activity of **A_5_** + *Xoo* treatment was not significantly increased. Therefore, the title compound **A_5_** might enhance rice resistance by increasing CAT, POD and PAL activities in rice.

### 2.9. Molecular Docking Study

To investigate the binding behavior of compounds in proteases, we performed molecular docking experiments. As shown in Figure 12, compound **A_5_** (A) and Bismerthiazol (B) have similar binding behaviors in proteins. For example, compound **A_5_** formed hydrogen bonds (3.3 Å and 3.4 Å) with the carboxyl groups of Ala35 and Cys153, respectively, with an interaction distance or bond length similar to that of Bismerthiazol (B). In addition, compound **A_5_** has 2 rings and is aromatic and has interactions with other amino acids due to its similar structure to Bismerthiazol, forming π-cation hydrophobic interactions with the carbonyl group of Ser34 (5.6 Å) and the amine of Arg157 (4.4 Å), respectively. This is something that Bismerthiazol does not possess, indicating that the introduction of aromatic structures in the compounds can contribute to the interaction with proteins and thus improve the inhibitory activity against bacteria. Moreover, the hydrophobic interactions formed are very important for the recognition and binding of compound **A_5_** to the protein.

### 2.10. Insecticidal Activity Analysis 

Bioassay results (Table 4) showed that most of the target compounds had good insecticidal activity against *P. xylostella*, among which compounds **A_3_ A_8_**, **A_11_**, **A_12_**, **A_22_**, **A_23_**, **A_24_**, **A_27_** and **A_33_** had the same activity as chlorantraniliprole at 500 μg/mL. The title compounds also showed good insecticidal activity against *Myzus persicae*, among which the activity of compounds **A_1_**, **A_3_** and **A_25_** was equivalent to that of commercial insecticide chlorantraniliprole at 500 μg/mL (Figure 13).

## 3. Materials and Methods

### 3.1. Physicochemical Properties Prediction

The computational assessment of the bioactivity score and the physicochemical properties of targeted structures was performed by using Molinspiration software (available online at www.molinspiration.com, accessed on 10 February 2023) with the structure-related calculation based on the SMILES file, which was generated from ChemDraw Ultra 8.0 [27,28,29,30].

### 3.2. Chemicals and Instruments

All reagent products from the Chinese Chemical Reagent Company were analytical or chemically pure. Thin-layer chromatography (TLC) of a GF254 silica gel pre-coated plate (Qingdao Haiyang Chemical Co., Ltd., Qingdao, China) was used to evaluate the progress of the reaction and the purity of the compounds. The ^1^H NMR, ^13^C NMR and ^19^F NMR spectra were determined in CDCl_3_ or deuterated dimethyl sulfoxide (DMSO-d_6_) solution using a JEOL-ECX-500 NMR (JEOL, Tokyo, Japan) or an AVANCE III HD 400M NMR (Bruker Corporation, Fallanden, Switzerland). TMS was added to the deuterated reagent as an internal standard. High-resolution mass spectrometry (HRMS) data were obtained on a Thermo Scientific Q Exactive (Thermo Scientific, Waltham, MA, USA). SEM measurements were performed on an FEI Talos F200C electron microscope with an acceleration voltage of 120 kV. All strains used *(Xoo*, *Xac*, *Psa* and Rs) were obtained from the laboratory of Guizhou university. The melting points of all title compounds were measured using an X-4 digital display microscopic melting point instrument (Henan Gongyi Yuhua Instruments Co., Ltd., Gongyi, China) without correction, and the yields of the compounds were not optimized. 

### 3.3. Synthetic Procedures of the Target Compounds

#### 3.3.1. Preparation of Intermediate 1 [31,32]

Thiourea (39.41 mmol) and potassium hydroxide were stirred in KOH (91.44 mmol) in 35 mL ethanol at 70 °C for 0.5 h. And then, ethyl acetoacetate (90.65 mmol) was slowly added to the mixture for 2 h. The reaction was concentrated under reduced pressure, the pH value was adjusted to 4, and the solid was collected by filtration and recrystallized with water to obtain intermediate 1, a white solid with a yield of 85–90%. Characterization data for intermediates are reported in the Appendix A.

#### 3.3.2. Preparation of Intermediate 2, 3 and 4 [33,34]

Intermediate 1 (7.03 mmol) and K_2_CO_3_ (10.55 mmol) were reacted with iodide methane (8.44 mmol), iodide ethane (8.44 mmol) and benzyl chloride (8.44 mmol) in 30 mL DMF to obtain intermediates 2, 3, and 4, respectively. Characterization data for intermediates are reported in the Appendix A.

#### 3.3.3. Preparation of Title Compound **A_1_**–**A_29_** [23,35]

Intermediate 2 (3.16 mmol) was added to a round-bottomed flask containing dichloromethane (25 mL) and triethylamine (9.48 mmol) and stirred for 30 min at room temperature; then, acyl chloride (3.79 mmol) containing different substituents was slowly added to the mixture. After 12 h of reaction, the solvent was removed under vacuum, and the title compound **A_1_**–**A_29_** was obtained by silica gel column chromatography (petroleum ether/ethyl acetate = 5:1) with a yield of 40–82%. Characterization data for final compounds are reported in the Appendix A.

#### 3.3.4. Preparation of Title Compound **A_30_**–**A_31_**

Intermediate 3 (2.94 mmol) was added to a round-bottomed flask containing dichloromethane (25 mL) and triethylamine (8.81 mmol), stirred at room temperature for 30 min, and then benzene sulfonyl chloride (3.52 mmol) was slowly added to the mixture for 12 h. When the reaction was over, the solvent was removed under a vacuum. The main compound **A_30_** was obtained by silica gel column chromatography (eluent: Petroleum ether/ethyl acetate = 5/1) the yield was 65%; The title compound **A_31_** was obtained in the same operation by replacing benzene sulfonyl chloride with pentafluorobenzene sulfonyl chloride with a yield of 69%. Characterization data for final compounds are reported in the Appendix A.

#### 3.3.5. Preparation of Title Compound **A_32_**–**A_33_**

Intermediate 4 (2.15 mmol) was added to a round-bottomed flask containing dichloromethane (25 mL) and triethylamine (6.46 mmol), stirred at room temperature for 30 min, and then slowly added benzene sulfonyl chloride (2.58 mmol) to the mixture for 12 h. The solvent was removed under a vacuum. The main compound **A_32_** was obtained by silica gel column chromatography (eluent: Petroleum ether/ethyl acetate = 5/1) the yield was 70%; The title compound **A_33_** was obtained in the same operation by replacing benzene sulfonyl chloride with pentafluorobenzene sulfonyl chloride with a yield of 75%. Characterization data for final compounds are reported in the Appendix A.

### 3.4. Crystallographic Analysis

A colorless single crystal of **A_10_** suitable for X-ray analysis was cultured from a mixture of N-hexane and ethyl acetate at room temperature. Single-crystal X-ray diffraction data were obtained on a Bruker Corporation diffractometer at 273.15 K using graphite monochromatic Cu Kα radiation (λ = 1.54178 Å). The structure was solved with the SHELXT structure solution program with intrinsic phasing and refined with the SHELXL refinement package by least squares minimization.

### 3.5. Antibacterial Activity Test

#### 3.5.1. Antibacterial Bioassay In Vitro

The inhibitory efficiency of title compounds on 4 types of bacteria in vitro was tested by the turbidimeter method at different concentrations [36,37,38]. For the initial screening of all 33 compounds, the solution concentration was set at 100 and 50 µg/mL, which was incubated with the bacterial solution and then procedurally measured for the OD value. A solution with no compound was set as a negative check, and BT, TZ and TC served as the positive control. Compounds that were active at this concentration were further tested at 5 lower gradient concentrations to get EC_50_. Data were collected in triplicate for each compound concentration. Based on the OD value, the inhibitory effect of the compound on bacteria was calculated. I (%) meant inhibition rate. CK meant the OD value of the non-drug control group. T meant the OD value of the test compound group.
I (%) = (CK − T)/T × 100

All tests were repeated 3 times for each compound. And the half maximal effective concentration (EC_50_) values were calculated with SPSS 21.0 software.

#### 3.5.2. Antibacterial Activity Bioassay In Vivo

We tested the curative and protective activities of compound **A_5_** against rice leaf blight in vivo under greenhouse conditions via leaf-cutting methods [39,40]. Commercial agents BT and TC were used as the positive control samples. In the following equation, CK and T mean the disease index for the negative control and treatment groups, respectively.
Control effect I (%) = (CK − T)/CK × 100

All of the experimental plants were grown in an environment of 30 ± 1 °C and 90% relative humidity, and each treatment had 3 replicates, and the biological activities data were calculated and analyzed with SPSS 21.0 software.

The bacteria inoculation method by leaf needlepoint modified from previous literature [41] was used to conduct the in vivo test of the title compound **A_33_** on citrus bacterial canker on citrus.

### 3.6. Mechanism Analysis

#### 3.6.1. Antibacterial Bioassay by Scanning Electron Microscopy

According to previously reported methods [42,43], the changes in the cell surface morphology after compound **A_5_** treatment were observed using SEM. The *Xoo* solution was washed 3 times and suspended in 1 mL of phosphate-buffered saline (PBS). Next, the DMSO solution containing compound **A_5_** was diluted to 100 and 50 μg/mL using PBS and incubated at 28 °C for 10 h. The test solution lacking the compound was used as a negative control. Cells were fixed with 2.5% glutaraldehyde for 8 h, and then anhydrous ethanol with different concentration gradients was used for dehydration. Finally, all the samples were observed with a Nova Nano SEM 450.

#### 3.6.2. The cytoplasmic Content Leakage Assay

The leakage of intracellular macromolecules was measured by determining the absorbance at 260 nm and 280 nm, and then the integrity of the cell membrane was evaluated according to the reported method [36,44,45]. The *Xoo* bacterial solution was cultured to OD_595_ = 0.6–0.8 at 28 °C for 180 rpm, rinsed with PBS 3 times and re-suspended in the same volume of PBS solution. Then **A_5_** (0, 21.2, 42.4, 63.6 and 84.8 μg/mL) was added. The absorbance of the solution was measured at 260 nm and 280 nm at different times (0, 12, 24, 36 and 48 h) after treatment with **A_5_** by a MicroplateReader (Bio-Tek Synergy2, Winooski, VT, USA).

#### 3.6.3. Test the Defense Enzyme Activity

Compound **A_5_** (200 μg/mL) was sprayed on rice plants that grew under greenhouse conditions for 30 days until droplets fell [46,47], and the rice plants were inoculated with *Xoo*. The plants were treated with TC and water in the same way as the positive and negative controls, respectively. Rice samples were collected 1, 3, 5 and 7 days after the bacterial infections. The activity of superoxide dismutase (SOD), peroxidase (POD), phenylalanine ammonia-lyase (PAL) and catalase (CAT) was tested using commercial ELISA kits following the manufacturer’s instructions (MEIMIAN, Yancheng, China).

### 3.7. Molecular Docking

A molecular docking station was built using the Ledock program according to the literature [48,49], the crystal structure of *Xoo*-Cas5d (PDB ID: 3VZI) was downloaded from Gallus gallus on the Protein Data Bank (https://www.pdb.org, accessed on 10 February 2023) and was processed with Pymol [50]. The molecular structures of compound **A_5_** and Bismerthiazol were drawn using ChemBioDraw Ultra 14.0 software and were optimized to minimize energy. A 17.5 × 15.3 × 14.7 docking box was generated with the Carboxin Standard in the protein as the center, and the docking station generated 20 ligand conformations with an RMSD less than 1.0 Å. The docking results were visualized in 3D by the Pymol software v.2.4.0 [51].

### 3.8. Insecticidal Assay

The insecticidal activity of compounds against *P. xylostella* and *Myzus persicae* was evaluated as follows [52,53,54,55]. All insecticidal assays were performed at (25 ± 1) °C using test insects raised in a laboratory and repeated according to statistical requirements. Insects were descended from different generations, and all remained unresistant. A filter paper was placed in a petri dish (inner diameter 9 cm), and water was added to increase humidity. On them, the fresh cabbage or tobacco leaves dics (6 cm in diameter) were placed. Those leaves were previously socked in a 500 ug/mL solution of compounds **A_1_** to **A_33_** for 10 s and air-dried. Then every 10 individuals of 3rd instar larvae of *P. xylostella* or *Myzus persicae* were subjected to the leaf in the Petri dish, and 3 repeats were set for each treatment. The death number were recorded every day, and mortality was calculated after 72 h. The solution with 0 ug/mL compound was set as the blank control, and the commercial chlorfenoacetamide served as the positive control. Each assay was conducted 3 times. The death rate was estimated by calculating the ratio of the number of deaths to the initial number of insects. The adjusted mortality rate (%) was as follows:

Adjusted mortality rate (%) = (death rate in the treatment group − death rate in the blank control group)/(1 − death rate in the blank control group) × 100.

## 4. Conclusions

In this study, we synthesized 33 novel pyrimidine derivatives with typical sulfonate patterns to explore innovative antibacterial frameworks. In vitro antibacterial test results showed that the target compounds showed good activity against four plants (Rs, *Psa*, *Xac* and *Xoo*) pathogens and insects (*P. xylostella* and *Myzus persicae*), and the optimal EC_50_ values for Xoo were 4.24 (**A_5_**), 6.77 (**A_31_**) and 9.35 (**A_33_**) μg/mL, respectively. The control effect of **A_5_** on rice bacterial leaf blight was superior to that of TC and BT; meanwhile, compound **A_33_** also exhibited a certain control effect on citrus canker leaf. The antibacterial mechanism suggested that these title compounds might cause shrinkage and rupture of pathogens by destroying the cell wall of pathogens *Xoo* and *Xac*. In addition, **A_5_** could increase the activity of related enzymes in rice and enhance its disease resistance. Based on the above promising results, these simple sulfonate structure pyrimidine derivatives can be further investigated as viable antibacterial products.

## Data Availability

All data generated in this study is presented in the current manuscript. No new datasets were generated. Data are available upon request from the corresponding author.

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
