# Peer review of "Design, Synthesis and Bioactivity of Novel Pyrimidine Sulfonate Esters Containing Thioether Moiety"

_ijms, 2023, doi:10.3390/ijms24054691_

Round 1

Reviewer 1 Report

The authors present an interesting study on the biological activity of a panel of compounds containing pyrimidine and sulfonate functionalities. The compound design follows a rational and logical progression from previous studies, and the biological properties assessed are relevant. The authors have performed a good job of characterising the compounds by 1H NMR, 13C NMR and mass specrometry. However, there are some issues with this manuscript that require addressing:

1) As the characterisation data is within the Supporting Information section, the authors could add a comment to Section 4.3 stating 'Characterisation data for final compounds and intermediates are reported in the Supporting Information', or something similar.

2) Lines 8089 require rewriting, as there are some incorrect statements here, and also there are no references cited. For example, the statement 'The results exhibited that all the 84 designed compounds were found in the zone for active drug molecule' does not make sense because the Lipinski rules are an indication of drug absorption and permeation properties, rather than drug activity.

The statement 'Lipinski's five rules for standard drug molecules are (milogP) (-0.4 ~ +5.6), (TPSA) (<160 A2), (Natoms) (20 ~ 70), (MW) (180 ~ 500 D), (nON) (nON ≤ 10), (nOHNH) (≤ 5), (Nviolations) (≤1), (Nrotb) (≤10), and (Volume) (≤500 A3)' needs to be corrected, because these are not actually the Lipinski rules (in fact there are only 4 Lipinski rules, the '5' refers to the fact that the values are multiples of 5. The Lipinski rule of fives is: No more than 5 H-bond donors, the molecular weight is not more than 500, the CLog P is not over 5 (or MLOGP is over 4.15) and the sum of N's and O's is not over 10.

3. The start of Section 1 and Section 2.1 contain the original text of the template, which should be deleted.

4. The authors may wish to consider Figures 1, 2 and 3 as a combined Figure 1a, b and c, as these are related to each other.

5. The crystallographic data could be moved to the supporting information.

6. The reader would benefit if the the data in Tables 3, 4 and 5 were represented graphically.

7. Reference citations should be '...[XX].' rather than '...x.'

8. Line 63: space missing (replacement(Figure3))

9. Line 281: comma missing

10. There should be a space between value and unit, e.g, line 286.

 Overall, I would recommend this manuscript for publication in IJMS after these changes have been made.

Author Response

Dear reviewer, it is an honor to have your approval of our work, and we are thankful for the reviewer’s time. On behalf of my co-authors, we would like to express our great appreciation to editor and reviewers. We have taken care of all the revisions accordingly.

R1: As the characterisation data is within the Supporting Information section, the authors could add a comment to Section 4.3 stating 'Characterisation data for final compounds and intermediates are reported in the Supporting Information', or something similar.

Responses: Thanks for your valuable comments and suggestions. This section has been modified in the manuscript according to your comments.

R1: Lines 80–89 require rewriting, as there are some incorrect statements here, and also there are no references cited. For example, the statement 'The results exhibited that all the 84 designed compounds were found in the zone for active drug molecule' does not make sense because the Lipinski rules are an indication of drug absorption and permeation properties, rather than drug activity.

The statement 'Lipinski's five rules for standard drug molecules are (milogP) (-0.4 ~ +5.6), (TPSA) (<160 A2), (Natoms) (20 ~ 70), (MW) (180 ~ 500 D), (nON) (nON ≤ 10), (nOHNH) (≤ 5), (Nviolations) (≤1), (Nrotb) (≤10), and (Volume) (≤500 A3)' needs to be corrected, because these are not actually the Lipinski rules (in fact there are only 4 Lipinski rules, the '5' refers to the fact that the values are multiples of 5. The Lipinski rule of fives is: No more than 5 H-bond donors, the molecular weight is not more than 500, the CLog P is not over 5 (or MLOGP is over 4.15) and the sum of N's and O's is not over 10.

Responses: Thanks for your valuable comments and suggestions. It has been modified in the manuscript according to your comments. “High bioavailability is more probable for a compound when there are ≤5 hydro-gen bond donors, ≤10 hydrogen bond acceptors, molecular weight ≤500, and Mi LogP ≤5; violation more than one of these rules may have problem with bioavailability. The structure of derivative 4-[2-amino-6-(4-substituted phe-nyl)-pyrimidin-4-yl]benzenesulfonamide (A1-A33) is predicted for their physicochemi-cal properties. The results exhibited that all the designed compounds were found in the zone for active drug molecule (Table 1). The Lipinski's five rules for standard drug molecules are (milogP) (-0.4 ~ +5.6), (TPSA) (<160 A2), (Natoms) (20 ~ 70), (MW) (180 ~ 500 D), (nON) (nON ≤ 10), (nOHNH) (≤ 5), (Nviolations) (≤1), (Nrotb) (≤10), and (Volume) (≤500 A3). The results of the physicochemical analysis showed that the de-signed compounds all conformed to Lipinski's rule of 5.” is changed to “The structure of derivative 4-[2-amino-6-(4-substituted phe-nyl)-pyrimidin-4-yl]benzenesulfonamide (A1-A33) was designed using ChemDraw Ul-tra 8.0 software and smile file was generated, that file was used to calculate the bioac-tivity score and physicochemical properties. The Lipinski's five rules for standard drug molecules is that no more than 5 H-bond donors, the molecular weight is not more than 500, the CLog P is not over 5 (or MLOGP is over 4.15) and the sum of N's and O's is not over 10. The results of the physicochemical analysis showed that the designed compounds all conformed to Lipinski's rule of 5(Table 1)[24,25]. The bioactivity score or drug-like score (A biological activity score of less than -5 indicates no activity, whereas a score greater than -5 indicates activity) corresponds to the drug active mol-ecule and the results show that the designed compounds (A1~A33) are located in the drug active molecule region (Table 2)[26].”

Table 2 The bioactivity score of target compounds A1-A33

Compd.

Bioactivity score

GPCR

ligand

Ion channel

modulator

Kinase

inhibitor

Nuclear receptor

ligand

Protease

inhibitor

Enzyme

inhibitor

A1

-0.41

-0.39

-0.64

-0.86

-0.47

-0.04

A2

-0.32

-0.44

-0.51

-0.72

-0.46

-0.06

A3

-0.34

-0.38

-0.53

-0.74

-0.44

-0.05

A4

-0.27

-0.36

-0.49

-0.67

-0.36

-0.02

A5

-0.22

-0.35

-0.42

-0.55

-0.21

0.10

A6

-0.35

-0.44

-0.62

-0.80

-0.43

-0.13

A7

-0.33

-0.38

-0.60

-0.78

-0.46

-0.06

A8

-0.28

-0.36

-0.40

-0.57

-0.32

0.02

A9

-0.23

-0.32

-0.45

-0.60

-0.38

-0.07

A10

-0.40

-0.40

-0.48

-0.75

-0.51

-0.12

A11

-0.49

-0.50

-0.66

-0.94

-0.54

-0.12

A12

-0.47

-0.46

-0.62

-0.90

-0.55

-0.12

A13

-0.45

-0.63

-0.51

-0.80

-0.58

-0.15

A14

-0.29

-0.40

-0.44

-0.58

-0.33

0.01

A15

-0.28

-0.36

-0.40

-0.57

-0.32

-0.02

A16

-0.44

-0.40

-0.62

-0.75

-0.47

-0.16

A17

-0.35

-0.44

-0.55

-0.70

-0.41

-0.08

A18

-0.16

-0.23

-0.37

-0.42

-0.22

-0.02

A19

-0.18

-0.25

-0.44

-0.46

-0.25

-0.01

A20

-0.45

-0.63

-0.86

-0.96

-0.51

-0.15

A21

-0.31

-0.46

-0.84

-0.76

-0.20

-0.02

A22

-0.52

-0.57

-0.95

-0.90

-0.49

-0.17

A23

-0.59

-0.61

-0.97

-0.93

-0.67

0.07

A24

-0.36

-0.51

-0.81

-0.76

-0.47

0.29

A25

-1.01

-0.84

-1.19

-1.02

-0.67

0.10

A26

-0.12

-0.31

-0.39

-0.50

-0.24

0.07

A27

-0.33

-0.40

-0.56

-0.75

-0.32

0.01

A28

-0.08

-0.24

-0.39

-0.54

-0.14

0.11

A29

-0.19

-0.32

-0.38

-0.57

-0.17

0.01

A30

-0.42

-0.47

-0.73

-0.89

-0.50

-0.09

A31

-0.27

-0.42

-0.54

-0.63

-0.28

0.04

A32

-0.25

-0.58

-0.47

-0.61

-0.22

-0.02

A33

-0.21

-0.52

-0.41

-0.52

-0.18

0.08

BT

-1.20

-1.27

-0.76

-1.50

-0.76

-0.20

TC

-0.95

-0.53

-1.10

-1.09

-0.97

-0.48

R1: 3. The start of Section 1 and Section 2.1 contain the original text of the template, which should be deleted.

Responses:  Thanks for your valuable comments and suggestions. It has been modified in the manuscript according to your comments.

R1: 4. The authors may wish to consider Figures 1, 2 and 3 as a combined Figure 1a, b and c, as these are related to each other.

Responses:  Thanks for your valuable comments and suggestions. It has been modified in the manuscript according to your comments.

R1: 5. The crystallographic data could be moved to the supporting information.

Responses:  Thanks for your valuable comments and suggestions. It has been modified in the manuscript according to your comments.

R1: 6. The reader would benefit if the the data in Tables 3, 4 and 5 were represented graphically.

Responses:  Thanks for your valuable comments and suggestions. It has been modified in the manuscript according to your comments.

R1: 7. Reference citations should be '...[XX].' rather than '...x.'

Responses:  Thanks for your valuable comments and suggestions. It has been modified in the manuscript according to your comments.

R1: 8. Line 63: space missing (replacement(Figure3))

Responses:  Thanks for your valuable comments and suggestions. It has been modified in the manuscript according to your comments.

R1: 9. Line 281: comma missing

Responses:  Thanks for your valuable comments and suggestions. It has been modified in the manuscript according to your comments.

R1: 10. There should be a space between value and unit, e.g, line 286.

Responses:  Thanks for your valuable comments and suggestions. It has been modified in the manuscript according to your comments.

Reviewer 2 Report

This is very useful and nice report, which deserves publication in IJMS.

However, I would like to suggest some minor corrections in the submitted manuscript prior to its final acceptation for publication.

1) Are the molecules and especially the best candidates are commercially available? If not, please indicate in "Conclusions" the time-scale of its/their availability. Are these synthetic molecules can be provided upon request to other colleagues for complemetary experiments for instance on their toxicity? A few words in Conclusions would also be useful on this subject. At last, are these molecules can be considered as hazardous? I mean if their use should be considered in secure conditions for users. If yes, it should also be mentioned somewhere  in the text .

2)  The suppression of a few unnecessary lines  is necessary. They probably belong to the "Instructions to Authors" generally found in the templates available on the journal website. (lines 22-27 in Section 1, Introduction, as well as lines 77-79 in Section 2.1, Physicochemical properties analysis).

Author Response

Dear reviewer, we are so appreciated your earnest review and great advice. We have taken care of all the revisions accordingly.

R2: 1) Are the molecules and especially the best candidates are commercially available? If not, please indicate in "Conclusions" the time-scale of its/their availability. Are these synthetic molecules can be provided upon request to other colleagues for complemetary experiments for instance on their toxicity? A few words in Conclusions would also be useful on this subject. At last, are these molecules can be considered as hazardous? I mean if their use should be considered in secure conditions for users. If yes, it should also be mentioned somewhere  in the text .

Responses:  Thanks for your valuable comments and suggestions. All compound molecules were synthesized and not commercially available. We plan to further study the related mechanism, and relevant experiments are in progress. It has been revised in the manuscript and thank you again for your valuable comments and suggestions.

R2: 2)  The suppression of a few unnecessary lines  is necessary. They probably belong to the "Instructions to Authors" generally found in the templates available on the journal website. (lines 22-27 in Section 1, Introduction, as well as lines 77-79 in Section 2.1, Physicochemical properties analysis).

Responses:  Thanks for your valuable comments and suggestions. It has been modified in the manuscript according to your comments.

Reviewer 3 Report

Jin, Xia and co-workers conducted quite a lot of synthetic work to yield a series of novel pyrimidine derivatives containing sulfonate groups. Moreover, their antibacterial and insecticidal activities were studied in detail. These extensive experiments implied simple sulfonate structure pyrimidine derivatives can be further investigated as new types of antibacterial and insecticidal products. These intriguing findings fits the scope of this journal, and could appeal extensive attentions from researchers. However, revisions are required if the article is considered for publication.

Comments:

1. All of the synthesized compounds were subjected to the antibacterial and insecticidal bioassays. Although these date were just listed in tables, they were lack of more in depth analysis, especially for the structure-activity-relationship, which could benefit for the development of new types of agents.

2. Did you detect the cytotoxicity of the compounds with potent antibacterial and insecticidal bioactivities? Because the pesticide were usually sprayed on the surface of plants.

3. ‘According to the structural analysis, A10 has a U-shaped conformation, which is close to its binding pose.’ The evidence such as molecular docking experiment was required for the above conclusion.

Others:

1. Please use the full names for the abbreviations when appeared for the first time, such as ‘CAT’.

2. Please delete the descriptive paragraphs ‘The introduction should briefly place the study in a broad context and...See the end of the document for further details on references.’ on P1L27–L35 and ‘’ on P3L77–79.

3. Please revise the styles of citations and of references according to the journal’s instructions.

4. Please move the ‘Conclusion’ part after the ‘Materials and Methods’ section.

Author Response

Dear reviewer, we are so appreciated your earnest review and great advice. We have taken care of all the revisions accordingly.

R3: 1. All of the synthesized compounds were subjected to the antibacterial and insecticidal bioassays. Although these date were just listed in tables, they were lack of more in depth analysis, especially for the structure-activity-relationship, which could benefit for the development of new types of agents.

Responses:  Thanks for your valuable comments and suggestions. It has been modified in the manuscript according to your comments. We have added the content of the SAR study in the manuscript, and that specific result is as follows.

Structure-activity relationship.

For the anti-Xoo activity, the aromatic ring (86%) was greater than pyridine ring (79%) than thiophene ring (75%) than aliphatic hydrocarbon (69%) than aliphatic hydrocarbon (45%) than fused ring (28%) under the same sulfonate structure. When both sulfonates were connected by aromatic rings, the activity of substituents at different positions was different, and the parapace substitution A3(83%) was higher than the collar substitution A2(73%). Activity is not the same, the same position of different substituents F (83%) > - NO2 (65%) > - Cl (62%), Br (62%) > - CN (58%) > - OCH3 (47%) > - CF3 (53%) > - C (CH3)3 (25%).

R3: 2. Did you detect the cytotoxicity of the compounds with potent antibacterial and insecticidal bioactivities? Because the pesticide were usually sprayed on the surface of plants.

Responses: Thanks for your valuable comments and suggestions. These experiments are ongoing, as we intend to conduct in-depth mechanistic studies of these compounds, so the relevant results are still under investigation. Thank you again for your helpful comments and suggestions.

R3: 3. ‘According to the structural analysis, A10 has a U-shaped conformation, which is close to its binding pose.’ The evidence such as molecular docking experiment was required for the above conclusion.

Responses: Thanks for your valuable comments and suggestions. It has been modified in the manuscript according to your comments. We add the evidence that the molecular docking results are in the manuscript. And that specific result is as follows.

2.9 Molecular Docking Study

To investigate the binding behavior of compounds in proteases, we performed molecular docking experiments. As shown in Figure X, compound A5 (A) and Bis-merthiazol (B) have similar binding behaviors in proteins. For example, compound A5 formed hydrogen bonds (3.3 Å and 3.4 Å) with the carboxyl groups of Ala35 and Cys153, respectively, with an interaction distance or bond length similar to that of Bismerthiazol (B). In addition, compound A5 has two rings and is aromatic and has interactions with other amino acids due to its similar structure to Bismerthiazol, forming π-cation hydrophobic interactions with the carbonyl group of Ser34 (5.6 Å) and the amine of Arg157 (4.4 Å), respectively. This is something that Bismerthiazol does not possess, indicating that the introduction of aromatic structures in the com-pounds can contribute to the interaction with proteins and thus improve the inhibitory activity against bacteria. Moreover, the hydrophobic interactions formed are very important for the recognition and binding of compound A5 to the protein.

Figure 12. Virtual molecular docking comparisons of title compound A5 (A) and Bismerthiazol (B) with Xoo-Cas5d (PDB code: 3VZI). Hydrogen bond interactions are indicated by a green line, and the π–π interactions are indicated by a purple line.

R3: 1. Please use the full names for the abbreviations when appeared for the first time, such as ‘CAT’.

Responses: Thanks for your valuable comments and suggestions. It has been modified in the manuscript according to your comments.

R3: 2. Please delete the descriptive paragraphs ‘The introduction should briefly place the study in a broad context and...See the end of the document for further details on references.’ on P1L27–L35 and ‘’ on P3L77–79.

Responses: Thanks for your valuable comments and suggestions. It has been modified in the manuscript according to your comments.

R3: 3. Please revise the styles of citations and of references according to the journal’s instructions.

Responses: Thanks for your valuable comments and suggestions. The format of the references in the manuscript has been completely revised according to your comments and is in full compliance with the literature format required by the journal.

R3: 4. Please move the ‘Conclusion’ part after the ‘Materials and Methods’ section.

Responses: Thanks for your valuable comments and suggestions. It has been modified in the manuscript according to your comments.
